# A Faster Training Algorithm for Regression Trees with Linear Leaves, and an Analysis of its Complexity

**Kuat Gazizov**
Dept. of Computer Science and Engineering
University of California, Merced
`kgazizov@ucmerced.edu`

**Miguel Á. Carreira-Perpiñán**
Dept. of Computer Science and Engineering
University of California, Merced
`mcarreira-perpinan@ucmerced.edu`

## Abstract

We consider the Tree Alternating Optimization (TAO) algorithm to train regression trees with linear predictors in the leaves. Unlike the traditional, greedy recursive partitioning algorithms such as CART, TAO guarantees a monotonic decrease of the objective function and results in smaller trees of much better accuracy. We modify the TAO algorithm so that it produces exactly the same result but is much faster, particularly for high input dimensionality or deep trees. The idea is based on the fact that, at each iteration of TAO, each leaf receives only a subset of the training instances. Thus, the optimization of the leaf model can be done exactly but faster by using the Sherman-Morrison-Woodbury formula. This has the unexpected advantage that, once a tree exceeds a critical depth, then making it deeper makes it faster to train, even though the tree is larger and has more parameters. Indeed, this can make learning a nonlinear model (the tree) asymptotically faster than a regular linear regression model. We analyze the corresponding computational complexity and verify the speedups experimentally in various datasets. The argument can be applied to other types of trees, whenever the optimization of a node can be computed in superlinear time of the number of instances.

## 1 Introduction

Decision trees have been a popular model in machine learning and statistics for many decades, having high interpretability and fast inference as major advantages, as well as very high accuracy when ensembled in forests. We consider trees making a hard decision at each node (rather than soft decision trees). We consider regression (rather than classification) trees, which define a mapping from $\mathbb{R}^D$ to $\mathbb{R}^E$ (where $E = 1$ is scalar regression), and we focus on oblique trees with linear leaves. Oblique trees use a (possibly sparse) hyperplane split at each decision node and are much more powerful than axis-aligned trees (which use a single-feature split) while remaining interpretable. (That said, all the results in this paper carry over to the axis-aligned case as well.) Linear leaves, which use a linear mapping $\mathbf{A}_i\mathbf{x} + \mathbf{b}_i$ as predictor, work much better for regression (i.e., continuous outputs) than constant leaves, which predict a constant output vector.

The traditional algorithms to learn regression trees, such as CART [3] or C5.0 and M5 [19, 20], are based on greedy recursive partitioning and have been used for decades. While they are fast, simple and intuitive, they do not optimize a global objective function of all the parameters in the tree, so they produce overly large trees and achieve low accuracy, particularly for oblique trees and/or for linear leaves. We focus instead on the family of *Tree Alternating Optimization (TAO)* algorithms, proposed more recently for classification [5], regression [28] and other tasks. TAO provides a proper optimization of a well-defined objective function, so that each iteration decreases the error or leaves it constant. Compared to CART or M5, this produces trees that are much smaller and much more accurate. The key idea of TAO, explained in detail for regression in section 3, is to optimize the

parameters of one node at a time in the tree, typically depthwise from bottom to top. One pass over all the nodes constitutes one TAO iteration, which is then repeated until convergence. This has important consequences: within each iteration, all nodes at the same depth can be optimized in parallel (*separability condition*); and the optimization of a single node (*reduced problem*) takes the form of a weight 0/1 loss binary classifier over a decision node, and of a linear regression fit over a leaf. The latter requires the solution of a linear system with a coefficient matrix of $D \times D$, where $D$ is the input dimensionality.

It turns out that this algorithm can be substantially accelerated while producing the same exact result. We give here an overview of the idea and full details in sections 3 and 4. We focus on the step over the leaves[1]. TAO has the peculiarity, unusual compared to most machine learning algorithms, that it never optimizes a model (here, leaf predictor) over the whole dataset of size $N$ instances. Instead, since the decision nodes make hard decisions and so the leaves partition the whole dataset into disjoint subsets, TAO fits independently a separate predictor $\{\mathbf{A}_i, \mathbf{b}_i\}$ for each leaf $i$ over only the instances currently reaching that leaf (its *reduced set*). Thus, the deeper the tree, the smaller the reduced set at each leaf; and, if the tree exceeds a critical depth $\Delta^* = \log_2 \frac{N}{D}$ and is complete (i.e., having $2^\Delta$ leaves for a depth $\Delta$) and balanced (i.e., having uniform reduced sets, each with $M = N2^{-\Delta}$ instances), then each leaf receives fewer instances than the number of features (dimensionality $D$). This makes the $D \times M$ data matrix $\mathbf{X}$ have rank at most $M$ (rather than $D$, as would typically be in practice). Thus, we can use the Sherman-Morrison-Woodbury formula [13, section 0.7.4] to solve the linear system exactly but inverting an $M \times M$ matrix (involving $\mathbf{X}^T\mathbf{X}$) rather than—as done in the original algorithm [28]—inverting a $D \times D$ matrix (involving $\mathbf{X}\mathbf{X}^T$). This results in a considerable speedup, particularly for high input dimensionality or deep trees.

The underlying principle here is that, if we have a problem whose cost on the sample size is $N^a$ for $a \in \mathbb{R}$, then partitioning the $N$ samples equally into $B$ disjoint groups and solving them has a total cost $N^a B^{1-a}$, i.e., $B^{1-a}$ times the cost without partitioning. If $a = 1$ this cost is independent of $B$, as happens with oblique decision nodes. But if $a > 1$ (superlinear cost), as happens with leaves with linear predictors, then solving for the partition is $B^{a-1}$ times faster. For the commonly occurring quadratic ($a = 2$) or cubic ($a = 3$) costs (as with solving linear systems) and when $B$ is large (note $B = 2^\Delta$ for a complete binary tree), the speedup can be huge.

This means the cost of the leaf step in our algorithm is not monotonically increasing with the depth (hence model size): it first increases until the critical depth $\Delta^*$, then decreases. This has the unexpected advantage that, once a tree exceeds the critical depth, then *making the tree deeper makes the leaf step faster to train*, even though the tree is larger and has more parameters. Indeed, *for deep enough trees and considering the total cost (decision nodes and leaves), this can make learning a nonlinear model (the tree) asymptotically faster than learning a regular linear regression model*.

The above discussion assumes a sequential updating of the nodes. However, in TAO all the nodes at the same depth define independent optimization problems, so they can be trained in parallel. Combined with the adaptive computation above, this makes the computational complexity much faster still.

What if the optimal tree size for a given dataset is not deep enough? In practice, fitting a model involves cross-validating its hyperparameters, usually by a grid search, which requires training trees of multiple depths. Thus, even if the final model selected is shallower than the critical depth (which may or may not be depending on the case), we do need to train deeper trees during the grid search, where our improved algorithm will speed up the training.

In summary, our paper has the following contributions. First, we propose an exact but faster leaf step, which makes our algorithm always better than the original one of [28]. This is described in section 3.1, after we review in detail the original algorithm in section 3. Second, in section 4, we analyze the computational complexity of our algorithm and show how the leaf step has a non-monotonic runtime across the depth domain (from depth 0, corresponding to a regular linear regression, through the critical depth, to depth $\log_2 N$, where each leaf has a single instance). We also point out that the idea applies to other types of trees, having decision nodes or leaves with superlinear cost on the sample size. Finally, in section 5, we verify the speedups experimentally (in the sequential compu-

---

[1]The idea applies to the decision nodes as well, but it makes no difference if the decision nodes have a computational cost $\Theta(ND)$ that is linear on the sample size, as is the case for oblique or axis-aligned trees, which we focus on here. However, if using decision nodes with a superlinear training cost on $N$, such as a kernel machine, the same non-monotonic cost would occur.

tation setting only) in various real-world datasets, where the reduced sets need not be balanced. We compare the original ("naive") algorithm of [28], which optimizes each leaf by solving a $D \times D$ linear system in the regular way (cubic cost on $D$); and our "adaptive" algorithm, which chooses the regular way or the SMW one depending on the leaf depth. We also apply the SMW formula to a traditional tree induction algorithm (CART) and show even larger speedups—as expected, since the trees it learns are much deeper than TAO's.

## 2 Related work

The literature on decision trees is vast. Here we focus on work on regression trees having linear predictors in the leaves. Decision trees date back to the 1950s [12, 21, 25], and typically use axis-aligned splits [3, 20] or oblique and sparse oblique splits [3, 5], with constant models at the leaves. Trees with linear leaves were first introduced in the M5 "model tree" [19] (further studied in [24, 25]), where a linear regressor is fit at each leaf after constructing a tree with constant labels using greedy recursive partitioning. The constant labels at the leaves are then replaced with linear models trained on the data reaching each leaf. Several variants of this idea have been proposed, including modifications to the splitting or pruning criteria [27, 17]. However, these methods remain fundamentally suboptimal for two main reasons: 1) the tree is constructed greedily by optimizing an impurity measure such as variance, and 2) the structure is learned under the assumption of constant outputs, ignoring the actual form of the final linear leaf models. As a result, the induced trees tend to be unnecessarily large and prone to overfitting [17]. More recently, exact optimization methods have been proposed using dynamic programming to construct globally optimal piecewise-linear regression trees [22]. While theoretically appealing, these methods do not scale to high-dimensional inputs or deep trees due to their combinatorial nature, and they rely on discretizing the input space via feature binarization. As such, they effectively solve a different problem from the original continuous formulation and are limited in practical applicability.

Optimizing decision trees is difficult because 1) the tree makes a hard decision at each node, which makes it a non-differentiable function, and gradient-based methods do not apply; and 2) there is a huge number of tree structures. Indeed, even in its simplest version (axis-aligned splits and binary inputs and outputs), the problem of training a decision tree is NP-hard [14]. The traditional greedy recursive partitioning procedure, such as CART [3] or C5.0 [20], is an approximate way to solve this problem, but it does not optimize an objective function of the entire tree's parameters (involving a well-defined loss function and regularization term, as with most modern machine learning models). This results in suboptimal and overly large trees. The Tree Alternating Optimization (TAO) algorithm [5] does optimize such an objective function and is scalable to large datasets. It assumes a given tree structure with parameters at the nodes and then optimizes the objective function over one node at a time in alternation, decreasing it at each step until convergence to a local optimum. The tree structure (e.g. its depth) can be selected by cross-validation. TAO produces trees that are both more accurate and smaller than greedy recursive partitioning [30]. The fact that TAO can optimize an essentially arbitrary choice of loss function and tree model has also made it possible to develop it for tasks to which trees have rarely been applied before, such as clustering [9], dimensionality reduction[2] [4] or semisupervised learning [29], as well as to optimizing forests [8, 6]. In this paper, we focus on the case of trees with linear leaves [28].

## 3 The original TAO algorithm for linear leaves and our speedup

The basic optimization algorithm for trees with linear leaves we use is the one in [28]. We give a self-contained description here. Our version is more general, allowing for a quadratic penalty on the linear regression weights (i.e., ridge regression). Later, in section 3.1, we describe how we modify it to speed it up.

Consider a fixed rooted directed tree structure consisting of decision nodes and leaf nodes, indexed by sets $\mathcal{D}$ and $\mathcal{L}$, respectively. The full set of nodes is $\mathcal{N} = \mathcal{D} \cup \mathcal{L}$. Each decision node routes an input instance $\mathbf{x} \in \mathbb{R}^D$ based on a decision function $f_i(\mathbf{x}; \boldsymbol{\theta}_i) \colon \mathbb{R}^D \to \mathcal{C}_i = \{\texttt{left}_i, \texttt{right}_i\}$. The

---

[2]In fact, the idea for the present work came from the tree autoencoder model [4], which first reported a non-monotonic cost for the leaf step as a function of the tree depth. There, each leaf of the tree contains a linear autoencoder (PCA), i.e., a low-rank linear regression.

decision function determines whether an instance is sent to the left or right child of decision node $i$. For oblique decision trees, if $\mathbf{w}_i^T\mathbf{x} + w_{i0} \geq 0$, the instance $\mathbf{x}$ is routed to the right child (i.e., $f_i(\mathbf{x};\boldsymbol{\theta}_i) = \text{right}_i$), and to the left otherwise, where $\boldsymbol{\theta}_i = \{\mathbf{w}_i, w_{i0}\}$. Leaf nodes predict outputs for instances within their decision region by applying $\mathbf{g}_i(\mathbf{x};\boldsymbol{\theta}_i)\colon \mathbb{R}^D \to \mathbb{R}^E$. In our case, we define a linear mapping $\mathbf{g}_i(\mathbf{x};\boldsymbol{\theta}_i) = \mathbf{A}_i\mathbf{x} + \mathbf{b}_i$, where $\boldsymbol{\theta}_i = \{\mathbf{A}_i, \mathbf{b}_i\}$, $\mathbf{A}_i \in \mathbb{R}^{E\times D}$ and $\mathbf{b}_i \in \mathbb{R}^E$. Finally, we define a tree mapping $\mathbf{T}(\mathbf{x};\boldsymbol{\Theta})\colon \mathbb{R}^D \to \mathbb{R}^E$, where $\boldsymbol{\Theta} = \{\boldsymbol{\theta}_i,\ i \in \mathcal{L}\cup\mathcal{D}\}$. The tree routes an input instance $\mathbf{x}$ along a unique path from the root to a leaf, where the corresponding leaf predictor is applied to produce the output.

To optimize the tree parameters, we define the following objective function for a given training set $\{(\mathbf{x}_n, \mathbf{y}_n)\}_{n=1}^N \subset \mathbb{R}^D \times \mathbb{R}^E$, where $L(\mathbf{y}, \mathbf{y}') = \|\mathbf{y} - \mathbf{y}'\|^2$ is the squared $\ell_2$-norm loss:

$$E(\boldsymbol{\Theta}) = \sum_{n=1}^N L(\mathbf{y}_n, \mathbf{T}(\mathbf{x}_n;\boldsymbol{\Theta})) + \lambda\sum_{i\in\mathcal{D}}\|\mathbf{w}_i\|_1 + \alpha\sum_{j\in\mathcal{L}}\|\mathbf{A}_j\|^2 \tag{1}$$

(throughout, all norms are Frobenius norms unless otherwise indicated). The regularization parameters $\lambda \geq 0$ and $\alpha > 0$ control the $\ell_1$ penalty for decision node parameters $\mathbf{w}_i$ and the $\ell_2$ penalty for leaf node parameters $\mathbf{A}_j$, respectively. We also define the *reduced set* $\mathcal{R}_i \subset \{1, \ldots, N\}$ of a node $i \in \mathcal{N}$ as the set of training instances that reach $i$ given the current tree parameters.

The algorithm is based on three key theorems, which we briefly discuss below. Detailed proofs can be found in [5]. We assume that the objective function $E(\boldsymbol{\Theta})$ is separable over training points and that parameters $\boldsymbol{\theta}_i$ are not shared across nodes.

**Theorem 3.1** (Separability condition). *Let $\mathbf{T}(\mathbf{x};\boldsymbol{\Theta})$ be the regression tree mapping, and $i, j \in \mathcal{N}$ be two nodes that are not descendants of each other. Denote the parameters in node $i$ and $j$ by $\boldsymbol{\theta}_i$ and $\boldsymbol{\theta}_j$, respectively. Let all other node parameters $\boldsymbol{\Theta}_{rest} = \boldsymbol{\Theta}\setminus\{\boldsymbol{\theta}_i, \boldsymbol{\theta}_j\}$ be fixed. Then the function $E(\boldsymbol{\Theta})$ of eq. (1) can be equivalently rewritten as*

$$E(\boldsymbol{\Theta}) = E_i(\boldsymbol{\theta}_i, \boldsymbol{\Theta}_{rest}) + E_j(\boldsymbol{\theta}_j, \boldsymbol{\Theta}_{rest}) + E_{rest}(\boldsymbol{\Theta}_{rest}) \tag{2}$$

*where $E_i$ does not depend on $\boldsymbol{\theta}_j$, $E_j$ does not depend on $\boldsymbol{\theta}_i$, and $E_{rest}$ is independent of $\boldsymbol{\theta}_i$ and $\boldsymbol{\theta}_j$.*

The main idea of this separability condition is that the reduced sets and weights of non descendant nodes are disjoint. As a result, each pair of non descendant nodes can optimize parameters independently using only their respective reduced sets.

**Theorem 3.2** (Reduced problem over a decision node). *Consider the objective function $E(\boldsymbol{\Theta})$ from eq. (1) and a decision node $i \in \mathcal{D}$. Assuming that all other node parameters remain fixed, the optimization problem $\min_{\mathbf{w}_i, w_{i0}} E(\boldsymbol{\Theta})$ can be reformulated in the following way:*

$$\min_{\mathbf{w}_i, w_{i0}} \sum_{n\in\mathcal{R}_i} \overline{L}_{in}(\overline{y}_{in}, f_i(\mathbf{x}_n;\mathbf{w}_i, w_{i0})) + \lambda\|\mathbf{w}_i\|_1 \tag{3}$$

*where: $\mathcal{R}_i$ is the reduced set of node $i$; we define the weighted 0/1 loss $\overline{L}_{in}(\overline{y}_{in}, \cdot)\colon \mathcal{C}_i \to \mathbb{R}^+ \cup \{0\}$ for instance $\mathbf{x}_n \in \mathcal{R}_i$ as $\overline{L}_{in}(\overline{y}_{in}, y) = l_{in}(y) - l_{in}(\overline{y}_{in})\ \forall y \in \mathcal{C}_i$, where $l_{in}(z) = \|\mathbf{y}_n - \mathbf{T}_z(\mathbf{x}_n;\boldsymbol{\Theta}_z)\|^2$ and $\mathbf{T}_z(\cdot;\boldsymbol{\Theta}_z)$ is the predictive function for the subtree rooted at node $z$; and we define the pseudolabel $\overline{y}_{in}$ as the child of $i$ that routes $\mathbf{x}_n$ to the the leaf with the lowest squared $\ell_2$-norm loss (in the case of a tie, where the losses for both children are equal for an instance $\mathbf{x}_n$, the instance $\mathbf{x}_n$ can be removed from the reduced problem, as its routing does not affect the loss).*

The pseudolabel $\overline{y}_{in}$ represents the preferred child for instance $\mathbf{x}_n$, determined by comparing the two losses incurred when routing it to the left and right child nodes and selecting the child with smaller loss. The weight assigned to $\mathbf{x}_n$ is the absolute difference between these losses, defined as $\gamma_{in} = |l_{in}(\text{left}_i) - l_{in}(\text{right}_i)|$. Thus, this results in a weighted 0/1-loss binary classification problem. Since optimizing this is NP-hard, we obtain an approximate solution by instead minimizing an $\ell_1$-regularized surrogate loss (we use logistic regression).

**Theorem 3.3** (Reduced problem over a leaf node). *Consider the objective function $E(\boldsymbol{\Theta})$ of eq. (1) and a leaf node $i \in \mathcal{L}$. Assuming that all other node parameters remain fixed, the optimization problem $\min_{\mathbf{A}_i, \mathbf{b}_i} E(\boldsymbol{\Theta})$ can be reformulated in the following way:*

$$\min_{\mathbf{A}_i, \mathbf{b}_i} \sum_{n\in\mathcal{R}_i} \|\mathbf{y}_n - (\mathbf{A}_i\mathbf{x}_n + \mathbf{b}_i)\|^2 + \alpha\|\mathbf{A}_i\|^2. \tag{4}$$

*The solution to this ridge regression problem is given by the following closed-form expression:*

$$\mathbf{A}_i^* = \mathbf{C}\mathbf{K}^{-1}, \qquad \mathbf{b}_i^* = \overline{\mathbf{y}}_i - \mathbf{A}_i^*\overline{\mathbf{x}}_i,$$

*where $\overline{\mathbf{x}}_i = \frac{1}{|\mathcal{R}_i|}\sum_{n \in \mathcal{R}_i} \mathbf{x}_n$, $\overline{\mathbf{y}}_i = \frac{1}{|\mathcal{R}_i|}\sum_{n \in \mathcal{R}_i} \mathbf{y}_n$, $\mathbf{C} = \sum_{n \in \mathcal{R}_i}(\mathbf{y}_n - \overline{\mathbf{y}}_i)(\mathbf{x}_n - \overline{\mathbf{x}}_i)^T$ is the cross-covariance matrix and $\mathbf{K} = \sum_{n \in \mathcal{R}_i}(\mathbf{x}_n - \overline{\mathbf{x}}_i)(\mathbf{x}_n - \overline{\mathbf{x}}_i)^T + \alpha\mathbf{I}_D$ is the regularized covariance matrix of the input, with regularization parameter $\alpha$ ($\mathbf{I}_D$ is the identity matrix of size $D$).*

The algorithm pseudocode is presented in the appendix. Despite the algorithm's complexity, it can be summarized as an iterative algorithm that optimizes tree nodes in a depthwise fashion until convergence. Each node is trained on its corresponding reduced set. Decision nodes learn to route training instances to the leaves where they achieve the lowest error, while leaves are trained as standard regression models. Each node update decreases the objective function or leaves it unchanged (if the approximate solution from the surrogate loss increases the objective, we skip the update). The hyperparameter $\lambda$ can make $\|\mathbf{w}_i\| = \mathbf{0}$ for a decision node $i$, effectively pruning one of its children, which results in TAO learning the tree structure (subject to being contained in the initial tree).

### 3.1 Fast solution of the leaf step using the SMW formula

Since the training set is partitioned over the tree leaves, each leaf receives only a subset of the training instances. If a subset size is smaller than the input dimension, we can solve its leaf linear system exactly but much faster using the Sherman-Morrison-Woodbury (SMW) formula [13, section 0.7.4]. This states that if $\mathbf{H} \in \mathbb{R}^{n \times n}$, $\mathbf{U} \in \mathbb{R}^{n \times r}$, $\mathbf{V} \in \mathbb{R}^{r \times n}$, and $\mathbf{J} \in \mathbb{R}^{r \times r}$ are given matrices such that $\mathbf{H}$, $\mathbf{J}$ and $\mathbf{J}^{-1} + \mathbf{V}\mathbf{H}^{-1}\mathbf{U}$ are nonsingular, then the inverse of $\mathbf{H} + \mathbf{U}\mathbf{J}\mathbf{V}$ can be computed as

$$(\mathbf{H} + \mathbf{U}\mathbf{J}\mathbf{V})^{-1} = \mathbf{H}^{-1} - \mathbf{H}^{-1}\mathbf{U}\left(\mathbf{J}^{-1} + \mathbf{V}\mathbf{H}^{-1}\mathbf{U}\right)^{-1}\mathbf{V}\mathbf{H}^{-1}.$$

In our case, assuming both matrices data $\mathbf{X}$, $\mathbf{Y}$ are centered, we need to solve a linear system involving the regularized covariance matrix $\mathbf{X}\mathbf{X}^T + \alpha\mathbf{I}_D$ of $D \times D$, which has a computational cost of $\mathcal{O}(D^3)$. However, using the SMW formula (substituting $\mathbf{H} = \alpha\mathbf{I}_D$, $\mathbf{U} = \mathbf{X}$, $\mathbf{J} = \mathbf{I}_N$, $\mathbf{V} = \mathbf{X}^T$), we obtain an expression based on the regularized Gram matrix of $N \times N$:

$$(\mathbf{X}\mathbf{X}^T + \alpha\mathbf{I}_D)^{-1} = \alpha^{-1}\mathbf{I}_D - \alpha^{-1}\mathbf{X}\left(\alpha\mathbf{I}_N + \mathbf{X}^T\mathbf{X}\right)^{-1}\mathbf{X}^T.$$

This formulation implies that instead of solving a linear system of size $D \times D$, we can solve an $N \times N$ system. We select the method with the lower computational cost based on the relative sizes of $N$ and $D$ by applying the following formula at each leaf (where $\mathbf{X}$ represents its reduced set):

$$\mathbf{K}^{-1} = \begin{cases} (\mathbf{X}\mathbf{X}^T + \alpha\mathbf{I}_D)^{-1}, & D \leq N \\ \alpha^{-1}(\mathbf{I}_D - \mathbf{X}\left(\alpha\mathbf{I}_N + \mathbf{X}^T\mathbf{X}\right)^{-1}\mathbf{X}^T), & N < D. \end{cases} \tag{5}$$

The leaf weight is given by $\mathbf{A}^* = \mathbf{C}\mathbf{K}^{-1}$. (Computationally, we do not compute a matrix inverse explicitly, but solve its associated linear system, which is faster and more stable.) This works not just for TAO but also for the traditional greedy recursive partitioning algorithms CART and M5.

## 4 Analysis of the computational complexity

We now give the computational complexity of our accelerated TAO training algorithm for an oblique decision tree with linear leaves. The formulas are somewhat involved because the computations depend on the size of the leaves' reduced sets. Consider a complete binary decision tree of depth $\Delta$ with sparse oblique decision nodes. We assume a sparsity coefficient $s \in [0, 1]$, where $s = 0$ corresponds to a completely sparse weight vector $\mathbf{w}$, and $s = 1$ corresponds to a fully dense weight vector[3]. Let the total number of training instances be $N$, the feature dimension be $D$, and the output dimension be $E$. We assume that the training instances distribute uniformly over the leaves ("balanced case"), so each leaf contains $M = N/2^\Delta$ instances; this is the best-case runtime. The worst-case corresponds to all instances going to the same leaf. In our experiments we observe a wide variation in leaf reduced set sizes, depending on the dataset. We apply the algorithm in a bottom-up fashion: leaves are trained first, followed by their parent nodes, up to the root. Below, we provide a time complexity analysis for one iteration of this process for decision and leaf nodes.

---

[3]Although $s$ varies across decision nodes and during training (as some weights become zero or nonzero), we take it to mean the average proportion of nonzero weights across nodes and across TAO iterations. This is valid because $s$ appears linearly in the runtime formulas.

## 4.1 Leaf nodes

The separability condition in TAO allows all leaves to be optimized independently, as they are non-descendant to one another. Training each leaf requires solving a ridge regression problem, which has a computational cost of $\mathcal{O}(MD^2 + D^3 + D^2E)$. This cost accounts for computing $\mathbf{K}$ and solving the linear system. Since $\mathbf{K}$ is positive definite, we apply the Cholesky decomposition, with cost of $D^3/3$ flops [10, Section 4.2.5]. The total cost for $2^\Delta$ leaves is therefore $2^\Delta\mathcal{O}(MD^2 + D^3 + D^2E) = \mathcal{O}(ND^2 + 2^\Delta(D^3 + D^2E))$. The exponential dependency on the tree depth $\Delta$ appears impractical, but using simple linear algebra techniques, such as the Sherman-Morrison-Woodbury formula, can significantly reduce this complexity. Call $\Delta^* = \log_2(N/D)$ the *critical depth*.

We define two regimes for training each leaf based on the relationship between $M = N/2^\Delta$ and $D$: *deep regime*, if $M < D$, which occurs when $\Delta \geq \Delta^*$; and *shallow regime*, when $\Delta < \Delta^*$. The final cost for training all leaves is:

$$\text{Cost} = \begin{cases} \Theta\big(ND^2 + D^3 2^\Delta + D^2 \min(N, E2^\Delta) + NDE\big), & \Delta \leq \Delta^*, \\ \Theta\big(N^2 D 2^{-\Delta} + N^3 2^{-2\Delta} + N^2 2^{-\Delta} \min(N2^{-\Delta}, E) + N^2 E 2^{-\Delta} + NDE\big), & \Delta > \Delta^*. \end{cases}$$

These formulas, while correct, appear complicated to understand and also give the misleading impression of having large costs, because of the terms on $2^\Delta$, $D^3$, $N^3$, etc. However, by applying the inequality in each case ($\Delta \leq \Delta^*$ or $\Delta > \Delta^*$) it follows that *the runtime in both regimes is upper bounded by* $\Theta(ND^2 + NDE)$. Importantly, this shows that, across the whole range of depths from 0 to $\log_2 N$:

- We do not incur a large, cubic cost $\Theta(D^3)$ or $\Theta(D^2E)$ (repeated for each leaf, i.e., $2^\Delta$ times) by naively solving a linear system for the linear mapping matrix $\mathbf{A}_i$ of $E \times D$. *The cost is only proportional to $D(D + E)$, i.e., quadratic on the dimension.*
- We do not incur a large, cubic cost on $N$. *The cost is only linear on $N$.*

Two extreme cases arise when: 1) a leaf contains one instance (i.e., $\Delta = \log_2(N)$), which has complexity $\Theta(ND + NE + NDE)$; 2) the tree has depth zero, so the tree reduces to a single ridge regression model with complexity $\Theta(ND^2 + NDE + D^3 + D^2E)$. Thus, *a deep enough tree (which is a nonlinear regression model) has a lower asymptotic complexity than a regular linear regression*.

In summary, what we have here is a happy collusion of two factors. First, a cost for the leaf predictors that depends either superlinearly on the dimension $D$ (and $E$) and linearly on the sample size $N$, or vice versa, depending on how it is computed: solving the regular linear system or using the SMW formula. And second, an effective sample size $N2^{-\Delta}$ at each leaf that decreases (exponentially) with the tree depth. Thus, switching to the computation that is superlinear on $N$ (the SMW formula) when the tree is deep enough (i.e., when each leaf receives fewer instances than dimensions) makes the total cost over all the leaves faster—so much so that the cost even decreases with the depth.

As shown above, in the deep regime the SMW formula is always faster, but how much? This can be seen by comparing the shallow and deep regimes' costs: the speedup is faster the deeper the tree is, or the larger the dimensionality $D$ is (and thus the lower the critical depth is).

**The particular case of scalar regression ($E = 1$)** A common case of regression is when we predict a scalar value rather than a vector. In that case, the complexity simplifies as follows:

$$\text{Cost} = \begin{cases} \Theta(ND^2 + D^3 2^\Delta + ND), & \Delta \leq \Delta^*, \\ \Theta(N^2 D 2^{-\Delta} + N^3 2^{-2\Delta} + ND), & \Delta > \Delta^*. \end{cases}$$

For deep enough trees, the term $NDE$ becomes $ND$, so the algorithm has no quadratic complexity on the dimension anymore, while the regular linear regression still contains terms that are quadratic and cubic on the dimension.

## 4.2 Decision nodes

Decision nodes are trained level by level. For a single decision node, the training process involves two steps: first, constructing a reduced problem by sending each training instance to both child nodes and computing their respective losses. Second, solving the constructed reduced problem with a logistic regression. To derive the total cost of all decision nodes, note that at the same level, the

total number of training instances equals $N$, and the tree has $\Delta$ levels of decision nodes. Then, the cost for constructing reduced problems for all decision nodes is: $\Theta\left(\frac{1}{2}sND\Delta(\Delta-1) + NDE\Delta\right)$. For solving the logistic regression problems, assume that each solver requires $c$ iterations on average. The total cost of solving logistic regressions across all levels is $\Theta(cND\Delta)$. Thus, the total computational complexity for training decision nodes is $\Theta(ND\Delta^2 + ND(E + c)\Delta)$.

### 4.3 Overall complexity

Combining the costs for leaf and decision node training, the total cost per iteration is $\Theta(ND\Delta^2 + ND^2 + NDE + ND(E + c)\Delta)$. Thus, the overall cost is linear on the number of instances $N$ and quadratic on the dimensionality $D$ or $E$. Note that this is asymptotically faster than solving a single ridge regression, in spite of the increased complexity of the tree's training procedure.

**Training parallelization** At level $l \in \{0, \dots, \Delta\}$, each node contains $N/2^l$ training instances. Given $2^\Delta$ cores, we can process all nodes at each level in parallel. Consequently, constructing a reduced problem can be distributed across training instances, resulting in a $2^\Delta$ speedup. The training complexity of all logistic regressions in the decision nodes is upper-bounded by a geometric series, resulting in $\Theta(2ND\Delta)$, which has a $\Delta/2$ speedup over the sequential computation. For leaf training, each core is assigned exactly one leaf, reducing the training time by a factor of $2^\Delta$ over the sequential computation.

**Inference time** Inference refers to mapping an input instance $\mathbf{x} \in \mathbb{R}^D$ to its corresponding leaf and applying a leaf predictor to compute the output. This involves traversing a single root-to-leaf path of depth $\Delta$. Given that the decision nodes are sparse, the cost of this traversal is $\Theta(sD\Delta)$, where $s$ is the sparsity coefficient of the decision nodes. Once the input instance reaches leaf $j$, the leaf predictor computes the output as $\mathbf{A}_j\mathbf{x} + \mathbf{b}_j$, which has a cost of $\Theta(ED)$. Thus, the total inference cost is $\Theta(D(s\Delta + E))$.

**Model selection** In practice when training a model on real data, the final tree (with best generalization) may be shallow or deep, depending on the case. This will be determined by a model selection criterion, such as cross-validation, over the depth and other hyperparameters of the tree, which typically requires a grid search. Whether the grid search is directly over $\Delta$ or over the decision node sparsity hyperparameter $\lambda$, this will result in leaves whose depth spans a range of values both below and above the critical depth $\Delta^*$. The speedup of our algorithm will occur in the depths above $\Delta^*$.

One problem with very deep trees is that the number of leaves and hence of parameters (in the linear mappings) becomes very large (particularly if the output dimension $E$ is high). Also, the resulting tree would likely overfit. This can be solved by using low-rank linear mappings, which are particularly suitable if there are fewer samples than features. This is a new type of tree model that we are working on and will report elsewhere.

## 5 Experiments

We evaluate the speedup of our algorithm ("adaptive") over the original algorithm ("naive") of [28] in several settings and verify that the models and RMSE obtained are the same for both (up to small numerical differences). We focus on speeding up TAO and some experiments also do this for CART linear trees (although they have a far lower accuracy, as shown in [28]). We implement the algorithm in C++ using the Eigen library [11] for linear algebra operations and LIBLINEAR [7] for solving $\ell_1$-regularized logistic regressions. The oblique decision tree is randomly initialized: each scalar weight is sampled from a standard normal distribution, and biases are adjusted to ensure that training instances are uniformly distributed across all leaves. Although TAO is highly parallelizable, we do not conduct experiments with parallel computation (section 4.3 gives a brief discussion). For preprocessing, we use the Python library PIL with a bilinear resampling filter for resizing and the `ndimage` package for rotation. For the Infinite MNIST dataset [18], we generated 1M points.

### 5.1 Speedup as a function of sample size, depth and feature dimension

Fig. 1 compares the naive and adaptive TAO versions with an oblique decision tree with linear leaves trained on the MNIST dataset ($N = 60\,000$). We fix the regularization parameters to $\lambda = 1$ and

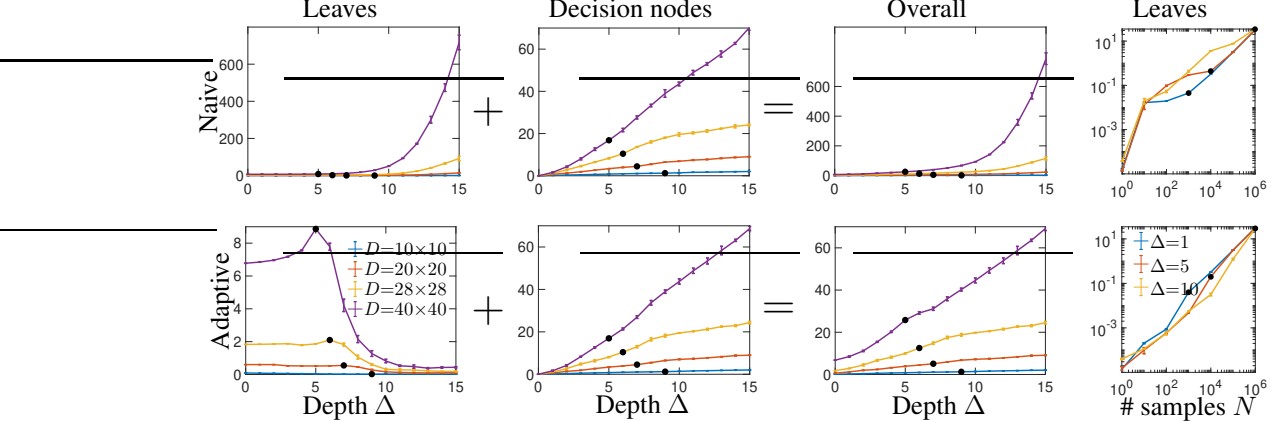

Figure 1: Training time (in seconds) for a single iteration on the MNIST dataset ($D = 784$) using the adaptive (top row) and naive (bottom row) algorithms for leaf training across different input dimensions $D$ and depths $\Delta$. Columns 1–3 show the training time for leaves, decision nodes and its total sum, respectively, as a function of $\Delta$. Critical depths $\Delta^*$ are indicated by black dots on each curve. Column 4 shows leaf training time as a function of the sample size $N$. Black dots now mark the critical sample size $N^*$ corresponding to each depth. Sample sizes smaller than or equal to the $N^*$ correspond to the deep regime, where the speedup occurs. Error bars indicate standard deviation over 5 runs.

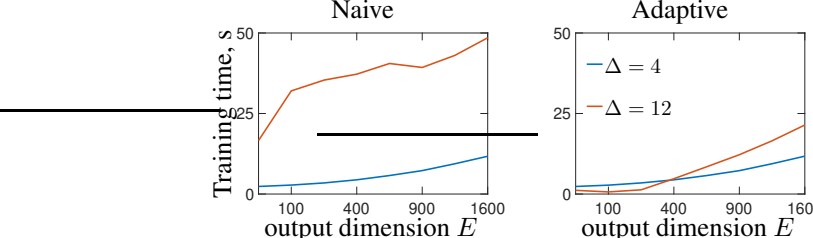

Figure 2: Training time (averaged over 5 runs) of all leaves for a single iteration on the MNIST dataset ($D = 784$) using the adaptive (left column) and naive (right column) algorithms for leaf training across different output dimensions $E$ and depths $\Delta$.

$\alpha = 0.001$, and construct a curve of training time over tree depth $\Delta$ for different input dimensions $D$. To generate datasets of different dimensions, we resize the MNIST images from their original size of $28 \times 28$ to $10 \times 10$, $20 \times 20$ and $40 \times 40$, resulting in 4 datasets with feature dimensions $D = \{784, 100, 400, 1600\}$ and corresponding critical depth $\Delta^* = \log_2(N/D) \approx \{6, 9, 7, 5\}$, respectively. Single-output regression labels are generated by applying a random projection unique to each class. Note that training at depth zero corresponds to solving a single ridge regression.

Starting from depth $\Delta = 10$, which corresponds to the *deep regime* for all datasets, training all leaves with the adaptive algorithm takes less than a second. For $D = 1600$, training all leaves at depth 10 is $8\times$ faster than training a single ridge regression model, and $16\times$ faster at depth 15. The training cost for all leaves has a non-monotonic behavior, decreasing after the critical depth. In contrast, the naive algorithm results in an exponential time complexity with respect to $\Delta$. This behavior is shown in fig. 1 (row 2), where leaf training dominates the overall training cost for deep trees, leading to an exponential total training time. Decision nodes, on the other hand, have similar training times for both the naive and adaptive algorithms, as they are unaffected by the specific leaf solver, and the time is linear over $N$.

In order to use larger datasets, we used 1M images of $28 \times 28$ from the Infinite MNIST dataset images; the input dimension $D = 784$ and output dimension $E = 1$ are fixed. Fig. 1 (column 4) demonstrates that the training time scales linearly with the number of samples for both the naive and the adaptive algorithms. The black dots mark the critical sample size $N^*$ for the given tree depth $\Delta$ and input dimension, i.e., $N^* = D2^\Delta$. As $\Delta$ increases, $N^*$ shifts to higher sample sizes, showing how the boundary between the deep and shallow regimes changes with depth. The training time curve lies in the deep regime for $N < N^*$ and in the shallow regime otherwise.

Fig. 2 shows the effect on the training time of the output dimension $E$. For both the naive and adaptive algorithms, the training time scales linearly with $E$ because both the leaf and decision node computations are linear in $E$. The critical depth here is $\Delta = 6$, so the red curve represents the deep

regime and the blue curve the shallow regime. Although both algorithms scale linearly with $E$, the naive one has a larger scaling factor of $\Theta(2^\Delta D^2)$, while for the adaptive one it is only $\Theta(ND)$. This indicates that while both methods experience increased training time as $E$ grows, the adaptive approach remains significantly more efficient in deep regimes.

## 5.2 Speedups on various real-world and synthetic datasets

Table 1 shows the training time of oblique decision trees on other datasets (whose descriptions are in the appendix), across shallow and deep regimes, using the naive and adaptive algorithms. All trees are trained with fixed hyperparameter values $\alpha = 0.01$ and $\lambda = 1$ for 20 iterations. The RMSE value (not shown) is identical for both algorithms. We present results ranging from small to large scale. The difference in training times becomes more pronounced for datasets with higher input dimension and larger sample size. For example, on moderate-sized datasets such as CT Slice, Rotated MNIST and Patched Fashion MNIST, the adaptive algorithm achieves up to a $3.5\times$ speedup over the naive one. On the datasets with high input dimension (e.g. $D = 2\,500$), the speedup reaches $25\times$.

Our complexity analysis in section 4 assumed a best case: complete trees (with $2^\Delta$ leaves for depth $\Delta$) with balanced leaves (having the same number of instances $N/2^\Delta$). However, our experiments are not restricted in that way. For example, in the TAO and CART linear trees of table 2, the number of leaves is much smaller than $2^\Delta$, indicating an irregular structure, and the reduced set sizes range from 8 to 5649 instances. Still, the adaptive algorithm shows a significant speedup.

## 5.3 Speedup on regression forests using bagging

The accuracy can be considerably increased if using an ensemble of trees (forest) rather than a single tree. Here we demonstrate this using bagging as ensembling mechanism, as in [28]. This means we use $T$ oblique trees, each trained independently on a bootstrap sample (or strict subset, by subsampling) of the original dataset, and define the forest output as the average of the $T$ trees. Our adaptive algorithm applies to each single tree, so the run-time gains (relative to the naive algorithm) are the same as if using a single tree. However, if using subsampling in bagging, each tree is trained on a subset having fewer than $N$ samples, so the critical depth becomes smaller, and the algorithm becomes faster.

Table 1 shows results for the Patched Fashion MNIST dataset using a forest of size $T = 30$ trees with depths ranging from 4 to 15. Each tree is trained for 20 iterations with a strict sample size of 90% of the training set.

Table 1: Training time (s) averaged over 5 runs for 20 iterations at different tree depths $\Delta$: adaptive vs naive algorithms.

| Dataset | $D$ | Impl. | $\Delta$ | T | Time (s) |
|---|---|---|---|---|---|
| ailerons | 40 | adap. | 13 | 1 | 7.67 |
|  |  | naive |  |  | 7.66 |
| CT slice | 384 | adap. | 15 | 1 | 119.12 |
|  |  | naive |  |  | 243.25 |
| sarcos | 21 | adap. | 16 | 1 | 34.72 |
|  |  | naive |  |  | 34.85 |
| Patched | 784 | adap. | 15 | 1 | 1160.24 |
|  |  | naive |  |  | 3990.65 |
| Rotated | 784 | adap. | 15 | 1 | 6611.74 |
|  |  | naive |  |  | 17534.83 |
| Random | 2500 | adap. | 15 | 1 | 3273.18 |
|  |  | naive |  |  | 83387.20 |
| Patched | 784 | adap. | 3 | 30 | 89.47 |
|  |  | naive |  |  | 89.06 |
| Patched | 784 | adap. | 7 | 30 | 217.58 |
|  |  | naive |  |  | 222.54 |
| Patched | 784 | adap. | 11 | 30 | 331.66 |
|  |  | naive |  |  | 515.36 |
| Patched | 784 | adap. | 15 | 30 | 580.12 |
|  |  | naive |  |  | 1995.32 |

## 5.4 Speedups across hyperparameter tuning

We evaluate the speedup in the practical setting of model selection by cross-validation, where to get the final model one needs to train multiple trees spanning a range of hyperparameter values and pick the best one. Since the adaptive algorithm does not change the learned trees, hyperparameter tuning and model selection is as with the naive one. We consider two methods of hyperparameter tuning.

**Grid-search hyperparameter optimization** We use a grid search over the regularization parameters $\lambda$ and $\alpha$ but not over the depth, which we set to relatively large (e.g. $\Delta = 15$). This is because varying $\lambda$ will automatically reduce the number of nodes from the initial, complete tree and produce a regularization path of trees of different number of leaves, each having a possibly different depth, and thus an irregular, learned tree structure. Thus, this grid search necessarily evaluates many trees of both deep and shallow regimes. In fact, when training trees of irregular structure, the idea of deep/shallow regime really applies to each individual leaf, which has its own depth. It is possible that the final selected model is shallow (i.e., $\Delta < \Delta^*$) and thus would not, for its final training, benefit from the adaptive speedup. However, this outcome cannot be known in advance. To find the optimal model, the grid search must evaluate the deeper trees (i.e., those with $\Delta > \Delta^*$), as the best

Table 2: *Left*: grid search time and final model training time on the Patched Fashion MNIST dataset. $\Delta$ denotes the depth of the final selected model, chosen from a regularization path that started at depth 15 for TAO and from a fully-grown tree for CART. *Right*: counts of sampled tree depths for Bayesian hyperparameter optimization frameworks Optuna and Hyperopt (100 trials each).

| | | Grid search | | | | | Bayesian hyperparameter optimization | | |
| Model | $\Delta$ | Train RMSE | Test RMSE | Search time, s | Training time, s | | Depth range | Optuna | Hyperopt |
|---|---|---|---|---|---|---|---|---|---|
| TAO-lin. adap. | 10 | 0.11 | 0.16 | 3485.38 | 253.82 | | 1–10 | 4 | 4 |
| TAO-lin. naive | 10 | 0.11 | 0.16 | 6899.90 | 256.05 | | 11–20 | 17 | 5 |
| CART-lin. adap. | 17 | 0.09 | 0.19 | 350.37 | 45.79 | | 21–30 | 64 | 62 |
| CART-lin. naive | 17 | 0.09 | 0.19 | 2444.19 | 54.08 | | 31–40 | 15 | 29 |
| Random Forest | 47 | 0.06 | 0.18 | N/A | 2734.23 | | | | |
| XGBoost | 6 | 0.11 | 0.17 | N/A | 2117.93 | | | | |

model may be in that regime. Our adaptive method provides a significant practical advantage by accelerating the evaluation of these deeper trees. Therefore, the total computational cost of the entire hyperparameter search is significantly reduced, as shown in table 2 (left). Note how the speedup for CART is much larger than for TAO; this is due to the fact that CART trees are much deeper and larger, in turn due to the suboptimality of CART as a tree learning method.

**Bayesian hyperparameter optimization**   A reviewer of this paper conjectured that using a more sophisticated hyperparameter optimization (HPO) method instead of a grid search, such as Bayesian hyperparameter tuning, might not sample deep trees often if they overfit, which might reduce the speedups achieved. To test this, we use Bayesian hyperparameter tuning for CART with linear leaves on the Patched Fashion MNIST dataset, using two HPO frameworks: Optuna [1] and Hyperopt [2]. Following standard practice for CART [12], we tune the cost-complexity pruning parameter. We run each framework for 100 trials with a time limit of 3 hours, minimizing the RMSE loss. The results in table 2 (left) demonstrate that our adaptive method is critical for the feasibility of these advanced HPO frameworks. The adaptive algorithm completed the full 100-trial search in under an hour. In contrast, the naive algorithm timed out after 3 hours, completing only 28% of the scheduled trials.

Furthermore, we analyze the distribution of the tree depths sampled by these frameworks to address the hypothesis that they would avoid deep trees. The results, shown in table 2 (right), demonstrate the opposite is true. Using Optuna, 79 of 100 trials resulted in trees with a depth greater than 20. The trend was even more pronounced with Hyperopt, where 91 of 100 trials resulted in trees with depths greater than 20.

This provides strong empirical evidence that Bayesian HPO frameworks do not avoid deep trees; on the contrary, they must extensively explore complex, deep structures to find the optimal model. This finding underscores that our adaptive speedup is not just beneficial for grid search, but is in fact essential for making modern HPO techniques computationally feasible.

## 6   Conclusion

We have improved the TAO algorithm for trees with linear leaves by making it considerably faster while producing the exact same result. We have achieved this by noting that the optimization over the leaf predictors separates over disjoint subsets of the training instances. The corresponding linear system can be solved either in the usual way, with a cost cubic on the input feature dimension, or via the Sherman-Morrison-Woodbury formula, with a cost cubic on the number of instances. Switching to the latter once the tree exceeds a critical depth gives a much faster algorithm. The idea applies also to the traditional CART or M5 algorithms, and to forests with linear leaves.

The argument can be applied to other types of trees, whenever the optimization of a node (whether a leaf or a decision node) can be computed in superlinear time of the number of instances. We are exploring this, particularly with low-rank linear mappings, which are more parsimonious than regular linear mappings.

## Acknowledgments and Disclosure of Funding

Work supported by NSF award IIS–2007147.

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

# A Algorithm

The algorithm is an iterative procedure that optimizes tree nodes in a depthwise fashion until convergence. Each node $i$ is trained on its corresponding reduced set $\mathcal{R}_i \subset \{1, \ldots, N\}$. Decision nodes learn to route training samples to the leaves where they achieve the lowest error, while leaves are trained as standard regression models. The pseudocode for the modified TAO algorithm is provided in fig. 3.

---

**input** training set $\{\mathbf{x}_n\}_{n=1}^N$;
      regularization hyperparameters $\lambda \geq 0$, $\alpha \geq 0$;
      initial tree $\mathbf{T}(\cdot; \boldsymbol{\Theta})$ of depth $\Delta$,
      where $\boldsymbol{\Theta} = \{\mathbf{w}_i, w_{i0}\}_{i \in \mathcal{D}} \cup \{\mathbf{A}_j, \mathbf{b}_j\}_{j \in \mathcal{L}}$
$\mathcal{N}_0, \ldots, \mathcal{N}_\Delta \leftarrow$ nodes at depth $0, \ldots, \Delta$, respectively
for each $i \in \mathcal{N}$: generate $\mathcal{R}_i$ (instances that reach node $i$)
**repeat**
  **for** $d = \Delta$ **down to** $0$
    **(par)for** $i \in \mathcal{N}_d$
      **if** $i \in \mathcal{L}$ **then**
        $\{\mathbf{A}_i, \mathbf{b}_i\} \leftarrow$ ridge regression solution
                  on $\mathcal{R}_i$ with penalty $\alpha$ (see Theorem 3.3 in the main paper)
      **else**
        for each instance $\mathbf{x}_n$ generate pseudolabels $\overline{y}_{in}$ and sample weights $\gamma_{in}$
        $\{\mathbf{w}_i, w_{i0}\} \leftarrow$ fit $\ell_1$-regularized weighted binary classifier on $\{(\mathbf{x}_n, \gamma_{in}, \overline{y}_{in})\}_{n \in \mathcal{R}_i}$
                  with penalty $\lambda$ (see Theorem 3.2 in the main paper)
    for each $i \in \mathcal{N}$: update $\mathcal{R}_i$ on tree $\mathbf{T}$
**until** stop
**return** $\mathbf{T}$

---

Figure 3: Pseudocode for the modified TAO algorithm with linear leaf optimization.

Table 3: Summary of datasets used in the experiments. $N$ denotes the number of samples, $D$ the input dimensionality, $E$ the output dimensionality, and $\Delta^*$ the critical depth.

| Dataset | $N$ | $D$ | $E$ | $\Delta^*$ |
|---|---|---|---|---|
| ailerons | 7154 | 40 | 1 | 7 |
| CT slice | 32100 | 384 | 1 | 6 |
| SARCOS | 44484 | 21 | 7 | 11 |
| Rotated MNIST | 60000 | 784 | 784 | 6 |
| Patched Fashion MNIST | 60000 | 784 | 64 | 6 |
| Random MNIST | 60000 | 784 | 1 | 6 |
| Infinite MNIST | 1000000 | 784 | 784 | 10 |

# B   Dataset description

Dataset characteristics are summarized in Table 3.

**ailerons** A dataset where the attributes describe the status of an aircraft. The task is to predict the control signal applied to the ailerons. `https://www.dcc.fc.up.pt/~ltorgo/Regression/DataSets.html`

**CT slice** A dataset from the UCI repository [16]. Each slice is represented by two polar histograms capturing bone structures and air inclusions, concatenated into a 384-dimensional feature vector. The target is the relative position of the slice along the axial axis. `https://archive.ics.uci.edu/datasets`

**SARCOS** A robotics dataset used in [23] where the task is to predict 7 joint torques from 21 input features including joint positions, velocities, and accelerations. `http://www.gaussianprocess.org/gpml/data/`

**Rotated** A regression task in which MNIST [15] digits are rotated by class-specific angles. The goal is to predict a rotated image from the original image. The rotation degrees are: 0: $8°$, 1: $49°$, 2: $-57°$, 3: $-63°$, 4: $16°$, 5: $-18°$, 6: $-10°$, 7: $-32°$, 8: $-71°$, 9: $58°$.

**Patched Fashion MNIST** A regression task where the model predicts an $8 \times 8$ patch extracted from a Fashion MNIST [26] image. The patch location depends on the digit label, with top-left corners positioned as follows: 0: $(17, 6)$, 1: $(10, 19)$, 2: $(17, 18)$, 3: $(6, 0)$, 4: $(2, 16)$, 5: $(13, 12)$, 6: $(10, 0)$, 7: $(0, 10)$, 8: $(11, 5)$, 9: $(11, 11)$.

**Random MNIST** A synthetic regression task where MNIST [15] digits are linearly mapped to a single scalar value using class-specific random projections. The goal is to predict this value from the original image.

**Infinite MNIST [18]** An augmented version of MNIST containing 1 million synthetically generated digits using pseudo-random deformations and translations. Each image is linearly mapped to a single scalar value using a class-specific projection, and the task is to predict this value. `https://leon.bottou.org/projects/infimnist`

# C Grid-search hyperparameter optimization

We provide an extended version of Table 2 from the main text, which also includes results for the CTSlice dataset.

Table 4: Hyperparameter search time and final model training time on the Patched Fashion MNIST and CTSlice datasets. $\Delta$ denotes the depth of the final selected model, chosen from a regularization path that started at depth 15 for TAO and from a fully-grown tree for CART-linear.

| Dataset | Model | $\Delta$ | Train RMSE | Test RMSE | Search time, s | Training time, s |
|---------|-------|----------|------------|-----------|----------------|------------------|
| Patched MNIST | TAO adap. | 10 | 0.11 | 0.16 | 3485.38 | 253.82 |
| | TAO naive | 10 | 0.11 | 0.16 | 6899.90 | 256.05 |
| | CART-lin. adap. | 17 | 0.09 | 0.19 | 350.37 | 45.79 |
| | CART-lin. naive | 17 | 0.09 | 0.19 | 2444.19 | 54.08 |
| CTSlice | TAO adap. | 9 | 0.44 | 1.51 | 548.92 | 62.65 |
| | TAO naive | 11 | 0.21 | 1.76 | 741.49 | 73.83 |
| | CART-lin. adap. | 31 | 0.14 | 2.75 | 116.90 | 4.60 |
| | CART-lin. naive | 31 | 0.14 | 2.69 | 1460.53 | 5.61 |

