# OpenReview forum: "A faster training algorithm for regression trees with linear leaves, and an analysis of its complexity"
_NeurIPS.cc/2025/Conference — NeurIPS 2025 poster_

### Official Review · Reviewer_U6L7 · 2025-06-08

**Clarity:** 3
**Significance:** 2
**Originality:** 3
**Rating:** 4
**Confidence:** 4

**Summary:**

The authors focus on learning oblique regression trees (with linear leaves). They propose a tweak to the TAO algorithm that speeds up training for deep trees. This tweak is exact and makes use of the fact that fitting a linear model at a leaf uses few samples, often making the rank of the data matrix constrained by the number of samples rather than the number of features. This yields a speedup, which the authors provide a complexity analysis for and confirm numerically.

**Questions:**

Can the authors provide more in-depth exploration of when the speed-up applies and how it relates to a model's predictive performance / overfitting?

Can the authors discuss in more detail how the method could be extended to other settings, e.g. learning sparse or axis-aligned splits? classification trees?

**Ethical Concerns:**

["NO or VERY MINOR ethics concerns only"]

**Final Justification:**

I thank the authors for their clarifications, but maintain my score of 4. I am still not convinced that a standard grid search will always involved deep enough trees to show a significant speedup, i.e. if a modest-depth tree overfits the data there is no need to search deeper trees.

**Limitations:**

The Limitations are not explicitly discussed in the Conclusion section, might be nice to discuss them more carefully there, esp how the speed-up relates to tree depths.

**Quality:**

3

**Strengths And Weaknesses:**

- S1: The authors propose an exact improvement to the TAO algorithm that can often provide a speed-up
- S2: The algorithm is intuitive, simple, and straightforward to implement
- S3: The experiments confirm the speed-ups, especially for deep trees
- W1: The impact of the speed-up is unclear, as it tends to kick in for large depths or limited-sample regimes where decision trees may tend to overfit. Speed & performance comparisons to other algorithms (including greedy methods like CART) would be helpful to show that there are realistic settings where this speed up applies & yields strong performance.
- W2: The introduction of the method and analysis of its complexity (Sec 3 & 4) are perhaps overly detailed for introducing the usage of the SMW formula, a fairly well-known trick
- W3: For a paper like this that focuses on an efficiency improvement, it would be nice to see code provided

---

> ### Author Rebuttal · Authors · 2025-07-31
>
> Thanks for your comments.
>
> **W1 and Q1.** We refer you to the reply "Q2. Hyperparameter tuning" and "Q1. Scope of Experiments" to reviewer 5aDC. There, we explain that a standard grid search 1) will estimate the tree with lowest validation error (as with any other type of model), and 2) will always involve deep enough trees where the speedup is significant (even if the selected tree is shallow). Thus, in practice there will always be a significant speedup over the grid search.
>
> **Q2**. Learning sparse splits: we already do this ($l_1$ term in eq. 1). Learning axis-aligned splits: this simply changes the reduced problem over a decision node to enumerating over features and thresholds (rather than fitting a linear classifier). Our speedup continues to happen in the leaves. Learning classification trees: we have not done this because it requires training a linear softmax classifier in each leaf; the SMW formula may still be useful there.

---

> > ### Comment · Reviewer_U6L7 · 2025-08-01
> >
> > I thank the authors for their clarifications, but maintain my score of 4. I am still not convinced that a standard grid search will always involved deep enough trees to show a significant speedup, i.e. if a modest-depth tree overfits the data there is no need to search deeper trees.

---

> > > ### Author Response · Authors · 2025-08-01
> > >
> > > We thank you for your follow-up. We would like to address your concern **”if a modest-depth tree overfits the data there is no need to search deeper trees”**  with the following use cases:
> > > - A key point is that a deep tree is not necessarily a complex tree. Tree complexity is more closely related to the number of leaves than to maximum depth. A tree can be very deep yet sparse (e.g., "lopsided"), having few leaves and representing a simple model. Conversely, a shallower, "bushy" tree can have many more leaves and severely overfit. The search space of trees with, say, max_depth=20 contains a much richer set of tree structures, including simpler ones, than the space of trees limited to max_depth=10. Stopping the search early risks missing these sparse, deep, and more accurate models. Indeed, we can see this effect directly in our experiments. As noted in our reply to reviewer 5aDC, on the Patched Fashion MNIST dataset, a tree of depth 15 might achieve 0% training error (perfect overfitting). However, the standard CART model finds a structure with a depth of 17 and 642 leaves. This provides clear evidence from the experiment that one must search beyond the point of initial overfitting to find better models.
> > >
> > > - The standard and popular CART algorithm grows a tree until the leaves are very small, resulting in an overfit tree. It then applies cost-complexity pruning to find an optimal subtree as described in [1]. This allows the algorithm to explore a large space of deep trees to find the best generalized model.
> > >
> > > - A critical use case where deep trees are required is in constructing Random Forests. The power of a Random Forest relies on averaging many diverse, low-bias models. To achieve this low bias, individual trees are intentionally grown to their maximum depth.
> > >
> > > - The grid search in TAO typically sets a large depth and then increases lambda from 0 to large in order to explore automatically those irregular structures and take advantage of warm-start in the optimization.
> > > - We have another hyperparameter ($\alpha$). A tree of depth Delta may overfit for some alpha values but not for others.
> > > - Cross-validation curves are noisy, so one doesn't stop as soon as the validation error increases.
> > >
> > > [1] Breiman, L., Friedman, J., Olshen, R.A., & Stone, C.J. (1984). Classification and Regression Trees (1st ed.). Chapman and Hall/CRC. https://doi.org/10.1201/9781315139470

---

### Official Review · Reviewer_5aDC · 2025-06-23

**Clarity:** 3
**Significance:** 3
**Originality:** 3
**Rating:** 4
**Confidence:** 4

**Summary:**

This paper proposes a faster training algorithm for oblique regression trees with linear predictors in the leaves, based on the Tree Alternating Optimization (TAO) framework. The authors modify the leaf-fitting step by applying the Sherman–Morrison–Woodbury (SMW) formula to invert an $N \times N$ matrix instead of a $D \times D$ one, where $N$ is the number of instances reaching a leaf and $D$ is the feature dimension. This results in an exact but significantly faster solution.

The key insight is that for deep trees exceeding a critical depth $\Delta^* = \log_2(N/D)$, each leaf receives fewer than $D$ samples, making SMW-based inversion more efficient. Surprisingly, making the tree deeper beyond $\Delta^*$ can reduce total training time, even though the model has more parameters.

The authors analyze the complexity and demonstrate substantial speedups (up to $25\times$) on high-dimensional ($D=2500$) regression tasks, with identical predictive performance to the original TAO algorithm.

**Questions:**

**1. Scope of Experiments:** Have you considered comparing your approach to other linear-tree algorithms, such as XGBoost or Random Forests? Even though those methods follow a different (greedy) strategy, it would be informative to see how TAO (with your speedup) compares in terms of speed and accuracy on benchmark tasks.
**2. Hyperparameter Tuning:** TAO requires setting the tree depth and regularization strength. Since deeper trees become “cheaper to train” in your approach, do you recommend always searching for depths beyond the critical threshold? Is there a risk of overfitting when training very deep trees? How do these hyperparameters influence performance in practice?
**3. Imbalanced Leaves:** The paper mainly analyzes balanced trees. How does the performance behave when the tree is highly unbalanced?

**Ethical Concerns:**

["NO or VERY MINOR ethics concerns only"]

**Final Justification:**

My concerns were partially addressed in the authors’ rebuttal, particularly regarding hyperparameter tuning (Q2) and imbalanced leaves (Q3). However, two major concerns remain, as noted in my official response: (1) the level of technical novelty, and (2) the breadth of the experimental study.

Given the improvements in the rebuttal, I am raising my score to 4. I am willing to raise it further, especially if other reviewers agree on the technical novelty of the contribution.

**Limitations:**

The paper does not explicitly discuss limitations. Some of the method’s limitations are mentioned above.

**Paper Formatting Concerns:**

No formatting issues noted.

**Quality:**

2

**Strengths And Weaknesses:**

**Strengths:**
1. The contributions of the paper are clearly stated (an exact faster leaf step, complexity analysis, experiments). The paper builds on the known TAO algorithm but identifies a nontrivial new insight that yields a significant practical gain.
2. Faster training of linear-model trees is useful for regression tasks, especially since TAO-trained trees can sometimes outperform greedy methods such as CART and M5 in accuracy. The proposed idea also applies to other tree-learning methods, as the authors note.
3. The use of the SMW identity is mathematically sound and yields exact equivalence to the original leaf solution. The derivations and complexity arguments are detailed and appear correct.
4. The paper provides a careful time-complexity analysis, covering both “shallow” and “deep” regimes, and clearly explains why a deep tree can be faster to train than a linear model.
5. The experiments on real and synthetic data demonstrate large speedups without loss of accuracy.


**Weaknesses:**
1. The main idea (use SMW for faster inversion) follows naturally from the observation that deeper trees have smaller leaf sample sizes. While the approach is effective, it builds on known mathematical tools and does not introduce a fundamentally new learning paradigm, but rather an optimization applied to TAO.
2. The paper lacks an explicit discussion on limitations. The proposed method inherits all the limitations of TAO. Namely, TAO optimizes a tree of given structure (depth and topology). The structure is initialized often as a full binary tree, and TAO only optimizes the parameters, not the structure. This leads to hyperparameter sensitivity, and poor initialization or overly deep/shallow structures can result in significant overfitting or degraded performance. Additionally, while TAO forests can be smaller overall, individual trees can be quite heavy (because both the splitting functions and the leaf models contribute substantial parameter counts). This is one reason why TAO training can be significantly slower than greedy methods such as CART or M5.
3. All experimental results focus only on TAO (adaptive) versus TAO (naive). No comparisons are shown against other linear-tree methods (e.g., CART, M5, or more recent PILOT [1]). The authors cite the TAO paper [2] to state that greedy trees have worse accuracy, but TAO is sometimes significantly slower. Since this paper accelerates TAO, it would be highly relevant to compare TAO (adaptive) to algorithms like XGBoost and Random Forests, as was done in the TAO paper [2].
4. The paper primarily discusses balanced trees. It would be helpful to discuss how the method performs on highly imbalanced splits.

**Reproducibility.** The authors do not mention whether they plan to release code or a library. No code archive is attached to this submission.

*[1] J. Raymaekers, P.J. Rousseeuw, T. Verdonck, and R. Yao. Fast linear model trees by PILOT. Machine Learning 113:6561–6610, 2024.*
*[2] A. Zharmagambetov, M. Carreira-Perpinan. Smaller, more accurate regression forests using tree alternating optimization. Proceedings of the 37th International Conference on Machine Learning, PMLR 119:11398–11408, 2020.*

---

> ### Author Rebuttal · Authors · 2025-07-31
>
> Thanks for your insightful questions.
>
> **Weakness 2.** Since our improved algorithm speeds up the training but does not change the learned tree, you are right that the limitations of linear trees and the original TAO algorithm do continue to apply; we'll note that in the paper. However, TAO does learn the structure to some extent, because (depending on the hyperparameter $\lambda$ in eq. (1), the weight vector of some decision nodes becomes zero and that node can be pruned (if $\lambda$ is large enough, the tree is pruned to a single leaf node). Thus, the resulting structure is a subset of the initial tree, and that structure is generally imbalanced. Also, you note that overly deep trees can result in significant overfitting; this is true, but the grid search will not select those.
>
> **Q2.** Hyperparameter tuning. This is one important point that we did note (in line 76 and 281), but is worth enlarging. Since our algorithm does not change the learned trees, hyperparameter tuning and model selection is as with the regular (naive) algorithm. We simply do a grid search over the hyperparameters; typically, we set the tree depth $\Delta$ to large enough (say, 15), vary the regularization hyperparameters $\lambda$ and $\alpha$, and pick the model with lowest validation error (when varying $\lambda$, the tree will progressively shrink, as noted above, automatically spanning a range of depths). As with any grid search, some of those trees will overfit and some will underfit, and the selected model may be shallow enough that the improved algorithm does not speed up that particular tree. But we don't know that until we do the grid search (in fact, it could be that the selected tree does show a large speedup). But, regardless of what the selected tree is, the grid search should span a range of depths so that we can corner the model with lowest validation error, and some of those depths (the larger ones, which have many linear leaves) will exhibit a large speedup. In short, the total training time (over the whole regularization path) is guaranteed to be significantly sped up, regardless of how shallow the selected model is. We illustrate this in the table below.
>
> **Q1.** Scope of Experiments. In the table below, we report other baselines you noted (XGBoost, RF, CART with linear leaves (M5)) and also the runtime for 1) the whole regularization path and 2) the selected tree (having lowest validation error).
>
> We conduct new experiments on the Patched Fashion MNIST dataset ($N=60000$, $D=784$, $E=64$, and the critical depth $\Delta^* = 6$) to provide the requested baseline comparisons and demonstrate the practical advantages of our method.
>
> *Experimental setup:* For the TAO and the CART baseline, we performed a hyperparameter search by constructing their respective regularization paths:
> - For TAO, we start with a complete balanced tree of depth 15 and increase the regularization parameter $\lambda \in [0.01, 1000]$, progressively pruning the tree at each step. We use warm-start, initializing each step from the previous tree.
> - For CART, we use the cost-complexity pruning path generated by scikit-learn's `cost_complexity_pruning_path` function.
>
> We compare our adaptive TAO against naive TAO, adaptive/naive CART-linear, Random Forest (default hyperparameters), and XGBoost (default hyperparameters). The reported time for CART models includes both initial tree construction and cost-complexity pruning.
>
> The table below shows the performance of the final selected model (chosen by lowest validation error) and the running time for both the whole regularization path and the final model training.
>
> Model | Final $\Delta$ | Train RMSE | Test RMSE | Number of leaves | Training time, s (whole regularization path) | Training time, s (selected tree)
> -|-|-|-|-|-|-
> TAO adaptive | 10 | 0.11 | 0.16 | 68 | 3485.38 | 253.82
> TAO naive | 10 | 0.11 | 0.16 | 64 | 6899.90 | 256.05
> CART-linear adaptive | 17 | 0.09 | 0.19 | 642 |350.37 | 45.79
> CART-linear naive | 17 | 0.09 | 0.19 | 642 | 2444.19 | 54.08
> Random Forest | 47 | 0.06 | 0.18 | 3791335 | | 2734.23
> XGBoost | 6 | 0.11 | 0.17 | 382313 | | 2117.93
>
> Note: For RF and XGBoost, "Final $\Delta$" is the maximum depth and "Number of leaves" is the total number of leaves across all trees in the ensemble.
>
> These experiments demonstrate:
> - TAO tree achieves the best test RMSE of 0.16, outperforming strong baselines like XGBoost and Random Forest.
> - The whole regularization path for the adaptive TAO is ~2x faster than the naive implementation. The speedup is even more pronounced for adaptive CART-linear, at ~7x faster. This confirms that our method provides a significant practical advantage by accelerating the entire model selection process, which must evaluate many deep trees, regardless of how deep the final selected model is.
>
> We will report similar experiments for the other datasets in the final paper.
>
> **Q3.** Imbalanced leaves. The trees learned by TAO and reported in our experiments (in the table above and in the paper) actually show significance imbalance. This is because 1) as noted above, the learned tree undergoes automatic pruning, so it is not complete, and 2) the reduced set of each leaf (set of training instances that reach it) need not be uniform even in a complete tree. For example, in the tree listed in the table above the reduced set sizes ranged from a maximum of 5649 to a minimum of 8 instances. Thus, although our complexity analysis considers the balanced case (which has the fastest runtime), the experiments show real-life results, which are far from balanced, yet the speedup is significant. We'll make this clear in the paper.

---

> > ### Comment · Reviewer_5aDC · 2025-08-06
> >
> > I appreciate the authors’ time and efforts in providing a detailed and constructive response to my review. My concerns were partially addressed, in particular Q2 (hyperparameter tuning) and Q3 (imbalanced leaves).
> >
> > That said, I still see two major concerns regarding this submission:
> > 1. **Technical Novelty** (raised by other reviewers as well). I acknowledge that in the original TAO paper the SMW formula was not used, and that sometimes even small tweaks can yield noticeable improvements and help popularize an algorithm. However, some reviewers feel that what the authors propose is a “fairly well-known trick” and may not be significant enough for a venue like NeurIPS. I am open to further discussion on this point.
> > 2. **Experimental Study**. The new experiment comparing TAO to XGBoost, Random Forest, and CART-linear on a single dataset (Patched Fashion MNIST) does strengthen the paper. However, I still find the experimental setup in the original TAO paper more appealing because: (1) it considers a broader set of datasets, and (2) it tunes hyperparameters for all baselines. I am not yet convinced that the observed ranking on Patched Fashion MNIST would hold more generally.
> >
> > I am raising my rating to 4 for now, but I remain open to discussion and am willing to raise my score further, especially if other reviewers agree on the technical novelty of the contribution.

---

> > > ### Author Response · Authors · 2025-08-08
> > >
> > > We thank the reviewer for increasing their score and for their participation in the discussion.
> > >
> > > **Experimental Study**
> > > To further demonstrate the generalizability of our findings, we extended our experiments with CTslice dataset ($N=53500$, $D=384$, $E=1$) using the same setup as in the Patched Fashion MNIST experiments (TAO adaptive/naive, CART-linear adaptive/naive, Random Forest, XGBoost).
> > >
> > > Model | $\Delta$ | Train RMSE | Test RMSE | Hyperparameter search time, s | Training time, s
> > > -|-|-|-|-|-
> > > TAO adaptive | 9 | 0.44 | 1.51 | 548.92 | 62.65
> > > TAO naive | 11 | 0.21 | 1.76 | 741.49 | 73.83
> > > CART-linear adaptive | 31 | 0.14 | 2.75 | 116.90 | 4.6
> > > CART-linear naive | 31 | 0.14 | 2.75 | 1460.53 | 5.61
> > > Random Forest | 69 | 0.06 | 1.46 | | 230.42
> > > XGBoost | 15 | 0.03 | 1.67 | | 105.50
> > >
> > > The results confirm the same qualitative conclusions:
> > > - adaptive TAO is faster over the full hyperparameter search (~1.35× speedup) while maintaining competitive accuracy.
> > > - CART-linear adaptive achieves a very large speedup (~12.58×) in the hyperparameter search compared to naive CART-linear.
> > >
> > > Due to time constraints before the discussion deadline, we cannot complete a full hyperparameter search for all baselines across every dataset. For the camera-ready version, we will 1) add similar experiments on additional datasets and 2) include full hyperparameter search for the ensemble baselines (Random Forest, XGBoost).

---

### Official Review · Reviewer_hbbP · 2025-07-01

**Clarity:** 3
**Significance:** 2
**Originality:** 2
**Rating:** 4
**Confidence:** 4

**Summary:**

The paper proposes an enhancement to the TAO (tree alternating optimization) algorithm for training regression trees with linear leaves. Specifically, they propose to optimize the leaves’ linear models using the Sherman-Morrison-Woodbury formula, taking advantage of the fact that each leaf only receives a potentially small subset of the full training set. On sufficiently deep trees, this has the potential to lead to significant speed ups, an observation that is verified experimentally across seven datasets.

**Questions:**

No specific questions, but please see weaknesses above.

**Ethical Concerns:**

["NO or VERY MINOR ethics concerns only"]

**Final Justification:**

Given the additional results provided by the authors in the discussion period that demonstrate the empirical gains of the proposed approach in a hyper-parameter tuning setting, I have increased my score. I still feel that the paper has limited technical novelty, as also noted by the other reviewers.

**Limitations:**

yes

**Quality:**

3

**Strengths And Weaknesses:**

Strengths:
- A simple improvement to speed-up the computation of linear regression models in the leaves of TAO regression trees
- The paper provides a clear analysis of the complexity of the proposed improvement
- Experiments show clear speed-ups over standard TAO

----

Weaknesses:
While the paper proposes a nice improvement that can speed up the computation of regression trees with high dimensionality, I found the paper to have limited novelty and limited significance.
- Limited technical novelty: TAO is well known algorithm Section 3 is a description of previous work primarily. The main change is the application of the SMW formula to solve the leaf problem faster in the “deep regime”.
- The experimental results show clear speed-up in performance, however significant gains are saved for very high-dimensional datasets with sufficiently deep trees such that the number of instances reaching a given leaf is smaller than the dimensionality. Gains in ensemble settings are more modest relatively.
- Deeper trees lead to reduced RMSE on training set, but do not necessarily generalize better as they may overfit. The experiments in the paper are not focused on generalization and investigation of whether the observed gains (that are focused on deep trees) are useful in practice.
- The work focuses on TAO regression trees exclusively. While the paper note that the main argument can be used to speed up greedy recursive partitioning algorithms such as CART, there is no experimental results demonstrating this. Further, the paper notes that this may extend to other types of trees but there is no additional information, examples, or experiments provided to substantiate this.
- Overall, while the contribution in the paper seems correct and potentially useful in some settings, it seems to me very incremental with relatively limited significance.

- minor:
* “Gram matrix” is used in Section 5 but is not previously defined.
* it would nice to have add y-axis labels to Fig 1 (especially as it seems the label for right-most figures is not the same as the rest?).

---

> ### Author Rebuttal · Authors · 2025-07-31
>
> Thanks for your insightful comments.
>
> **Limited novelty and limited significance**: our use of the SMW formula may seem obvious in hindsight, but: 1) the authors of the ICML 2020 paper did not realize this for the TAO algorithm; 2) the many researchers working since the 1980s on the case of CART linear trees did not realize either; and 3) as noted by reviewer 5aDC, we provide a "nontrivial new insight that yields a significant practical gain", since the speedup is large and doesn't involve any approximation, for both TAO or CART. We have added results to speed up CART linear trees in the table in the reply to reviewers 5aDC.
>
> Also, our paper has a second contribution: the non-monotonic behavior of the training complexity predicted by our analysis, very unusual in machine learning models. As reviewer 5aDC notes: "Surprisingly, making the tree deeper beyond can reduce total training time, even though the model has more parameters".
>
> **Significant gains are saved for very high-dimensional datasets with sufficiently deep trees [...] Deeper trees [...] may overfit.** This is one important point that we did note (in line 76 and 281), but is worth enlarging. Since our algorithm does not change the learned trees, hyperparameter tuning and model selection is as with the regular (naive) algorithm. We simply do a grid search over the hyperparameters; typically, we set the tree depth $\Delta$ to large enough (say, 15), vary the regularization hyperparameters $\lambda$ and $\alpha$, and pick the model with lowest validation error (when varying $\lambda$, the tree will progressively shrink, as noted above, automatically spanning a range of depths). As with any grid search, some of those trees will overfit and some will underfit, and the selected model may be shallow enough that the improved algorithm does not speed up that particular tree. But we don't know that until we do the grid search (in fact, it could be that the selected tree does show a large speedup). But, regardless of what the selected tree is, the grid search should span a range of depths so that we can corner the model with lowest validation error, and some of those depths (the larger ones, which have many linear leaves) will exhibit a large speedup. In short, the total training time (over the whole regularization path) is guaranteed to be significantly sped up, regardless of how shallow the selected model is. We illustrate this in the table in the response to reviewer 5aDC. This now shows the speedup for the selected tree (rather than a deep tree) and the for the whole regularization path (grid search). As you can see, a standard grid search 1) will estimate the tree with lowest validation error (as with any other type of model), and 2) will always involve deep enough trees where the speedup is significant (even if the selected tree is shallow). Thus, in practice there will always be a significant speedup over the grid search.

---

> > ### Comment · Reviewer_hbbP · 2025-08-02
> >
> > Thank you for your response. I acknowledge the authors' explanation about the speed up in the context of grid search where potentially unnecessarily deep trees are considered as part of the grid search even if they lead to worse performance. I would note that if that is the concern, moving from grid search to a more sophisticated procedure such as bayesian hyperparameter tuning should reduce this problem assuming the very deep trees are indeed not generalizing well. I think what is interesting to see is whether these deep trees, where the gains in terms of speed up are significant, tend to overfit and therefore will not be sampled often by a more sophisticated tuning procedure than grid search.

---

> > > ### Author Response · Authors · 2025-08-04
> > >
> > > Thanks for bringing up Bayesian hyperparameter tuning, which we had not considered (for the reasons explained below), and which actually gives more support to our speedup. We'll include this in the paper.
> > >
> > > Firstly, as to your statement **"whether these deep trees, where the gains in terms of speed up are significant, tend to overfit"**: this is an empirical question. We reiterate that in the example table we gave you the tree selected by cross-validation was above the critical depth for both TAO and (very much so) for CART, hence it is a "deep tree" that does not overfit.
> > >
> > > That said, we tried Bayesian hyperparameter tuning. We performed a hyperparameter tuning experiment for CART with linear leaves using two popular frameworks, Optuna and Hyperopt, on the Patched Fashion MNIST dataset.
> > >
> > > In this experiment, we tuned the cost-complexity pruning parameter by searching over all effective $\alpha$ generated by scikit-learn's `cost_complexity_pruning_path` function. For each framework, we compared the adaptive method against the standard naive approach, running for 100 trials and minimizing RMSE loss. A time limit of 3 hours was set for each run.
> > >
> > > The results demonstrate a noticeable difference between the adaptive and naive implementations. Our adaptive method completes the full 100-trial search in under an hour. In contrast, the naive approach fails to complete the search, timing out after 3 hours having finished only 28% of the required trials.
> > >
> > >
> > > Framework | Implementation | Progress | Time
> > > -|-|-|-
> > > Optuna | adaptive | 100/100 | 52m
> > > Optuna | naive | 28/100 | > 3h (timeout)
> > > Hyperopt | adaptive | 100/100 | 59m
> > > Hyperopt | naive | 28/100 | > 3h (timeout)
> > >
> > >
> > > We would like to directly address the reviewer's concern **"whether these deep trees [...] will not be sampled often by a more sophisticated tuning procedure."** Our findings show the opposite is true. We demonstrate the distribution of the sampled tree depths:
> > > Using Optuna, the search favored deep structures: 79 out of 100 trials (79%) resulted in trees with a depth greater than 20.
> > >
> > >
> > > Depth range | Count | Avg #leaves | min #leaves | max #leaves
> > > -|-|-|-|-
> > > 1-10  | 4  | 26    | 5     | 56
> > > 11-20 | 17 | 273   | 91    | 767
> > > 21-30 | 64 | 6578  | 1110  | 21141
> > > 31-40 | 15 | 35583 | 23232 | 49075
> > >
> > >
> > > This trend is even more pronounced with Hyperopt, where 91 out of 100 trials (91%) resulted in trees with depths greater than 20.
> > >
> > >
> > > Depth range | Count | Avg #leaves | min #leaves | max #leaves
> > > ------|----|-------|-------|------
> > > 1-10  | 4  | 41    | 15    | 75
> > > 11-20 | 5  | 252   | 92    | 679
> > > 21-30 | 62 | 10199 | 1082  | 21901
> > > 31-40 | 29 | 34502 | 23167 | 49503
> > >
> > >
> > > This provides strong quantitative evidence that the sophisticated hyperparameter optimization frameworks do not avoid deep trees. On the contrary, they extensively explore complex, deep structures to find the optimal model.
> > >
> > > **A further point:** Bayesian hyperparameter tuning for decision trees is computationally highly inefficient for the tree depth and structure, because one can quickly scan the subsets of a deep tree and cross-validate them. As we said in our earlier reply, a cross-validation grid search for trees proceeds from deep trees to shallow trees by searching over smaller subsets of those trees; the details vary between CART and TAO. With CART, this is the standard cost-complexity procedure: you grow a tree *in full*, then prune it using a validation set. (For Random Forests you do not even prune.) For TAO, you train a deep tree for $\lambda=0$ (no sparsity) and then increase lambda, so the pruning happens automatically. In both cases this is far more efficient than training trees from scratch: for CART, because of how the cost-complexity algorithm works (replace subtrees with leaves without touching anything else); for TAO, because the tree for a somewhat smaller lambda typically requires just one iteration thanks to the use of warm-start (this doesn't occur in the direction of decreasing lambdas because, as we said earlier, it is unclear how to "grow" the tree). This is simply how things are done in practice, at least at present, and they do imply one tests deep trees. Finally, a grid search can run in parallel if there are multiple hyperparameters (e.g. over alpha in our case).

---

> ### Comment · Reviewer_hbbP · 2025-08-04
>
> Thank you very much for your detailed response. This additional experiments are very interesting, however I am still confused about the experimental setting: did you set max_depth as a tunable parameter together with the typical regularization parameters for the tree construction (e.g., min_samples_split, min_samples_leaf, max_features, min_impurity_decrease, etc)?
> The main point is that if deep trees tend to overfit compared to shallower trees (when used in conjunction with the various standard tree regularization techniques) they should not be sampled often during the bayesian hyper-parameter search. If these experiments were done with a high max depth and only rely on tuning the cost complexity pruning parameter which is a post-pruning technique, then all experiments would first have to solve a very deep tree which will naturally benefit from the proposed technique due to this specific design of the experiment. Naturally, if the only thing tuned is the cost complexity parameter for ccp, a bayesian approach would not be useful (as the authors note) and you would go "from deep trees to shallow trees". However, my original point considers a hyper-parameter tuning that covers the widely used pre-pruning regularization techniques (max_depth, min_samples_leaf, min_impurity_decrease, etc). I would appreciate the authors' clarification on this.

---

> > ### Author Response · Authors · 2025-08-05
> >
> > We thank the reviewer for their feedback and active participation in this discussion.
> >
> > Our initial focus on tuning the cost-complexity pruning parameter (`ccp_alpha`) stems from post-pruning being a well-established and preferred strategy for training CART decision trees. For example, "The Elements of Statistical Learning" p. 308 states *"the optimal tree size should be adaptively chosen from the data [...] The preferred strategy is to grow a large tree, stopping the splitting process only when some minimum node size (say 5) is reached. Then this large tree is pruned using cost-complexity pruning."*.
> >
> > To directly address the reviewer's point about pre-pruning, we conducted a new experiment using the Optuna framework using 100 trials. In this search, we tuned `max_depth: [5, 40]`, `min_samples_leaf: [1, 1000]`, `max_features: [0.1, 1.0]`. This is summarized in the table below:
> >
> > Depth range | Count | Avg #leaves | min #leaves | max #leaves
> > -|-|-|-|-
> > 1-10  | 33 | 61 | 8 | 94
> > 11-20 | 66 | 134 | 65 | 610
> > 21-30 | 1 | 45054 | 45054 | 45054
> >
> > The depths that are favored by the HPO framework still correspond to the trees in deep regime (see the avg number of leaves).
> >
> > References:
> > Hastie, T., Tibshirani, R.,, Friedman, J. (2009). The elements of statistical learning: data mining, inference and prediction. Springer.

---

> > > ### Comment · Reviewer_hbbP · 2025-08-07
> > >
> > > Thank you for your response and for providing these additional results that demonstrate the empirical benefits of the approach. I will therefore increase my score.

---

> > > > ### Author Response · Authors · 2025-08-08
> > > >
> > > > We thank the reviewer for increasing their score and for the constructive comments and discussion. Please let us know if there are any further points that remain unaddressed.

---

### Official Review · Reviewer_UDF8 · 2025-07-03

**Clarity:** 3
**Significance:** 3
**Originality:** 2
**Rating:** 4
**Confidence:** 4

**Summary:**

This paper closely examines the Tree Alternating Optimization(TAO) algorithm for regression trees with linear leaves, which was introduced in an ICML 2020 paper [1]. TAO supposedly provides trees that are "much smaller and much more accurate" than those produced by other older popular algorithms like CART. This is believable and I did not verify the results from the ICML 2020 paper. TAO iterations consist of two types of steps, those optimizing the internal decision nodes of the tree, and those optimizing the leaf nodes. The leaf node optimization step is essentially the same as solving a ridge regression problem for the data points sent to that particular leaf. In the original paper introducing TAO, they perform a "naive" ridge regression whereas in this submission, they show that they can substantially speed up this step (at least for higher dimensionality of the data and/or higher tree depth) using the Sherman-Morrison-Woodbury formula. Basically, the trees produced by the algorithm provided in this paper are identical to those produced by TAO but they can compute them faster. The submission shows improved computational complexity by some basic analysis and furthermore they also experimentally show that their algorithm is faster than TAO.

References:
1) "Smaller, more accurate regression forests using tree alternating optimization" by A. Zharmagambetov and M. Á. Carreira-Perpiñán, ICML 2020

**Questions:**

Minor typos/clarifications:

Lines 54-58: I believe you want to mention $N'$ here instead of $N$ where $N'$ is the effective sample size $N2^{-\Delta}$, which you mention at Line 240. Since you are using $N$ for the size of the dataset in the immediately preceding lines in this paragraph from Lines 47-53, it gets a bit confusing here.

Line 56: but -> by

Line 113: peocedure -> procedure

**Ethical Concerns:**

["NO or VERY MINOR ethics concerns only"]

**Final Justification:**

I have considered the author's rebuttal regarding originality and also gone over the NeurIPS 2025 reviewer guidelines regarding originality where it is stated that: **As the questions above indicates, originality does not necessarily require introducing an entirely new method. Rather, a work that provides novel insights by evaluating existing methods, or demonstrates improved efficiency, fairness, etc. is also equally valuable.**

The paper clearly demonstrates improved efficiency and so I feel its major weakness of Originality that I held previously is not so strong. But I still feel that it is a borderline paper and so I am only increasing my score to Borderline accept from my earlier score of Borderline reject.

**Limitations:**

Yes

**Quality:**

4

**Strengths And Weaknesses:**

Quality: The submission is technically sound with claims that are well-supported by both theoretical results and experimental validation. The methods used are appropriate and this is a complete piece of work.

Clarity: The submission is overall clearly written with minor typos which I list in the *Questions* section of the review.

Significance: The submission is certainly relevant to NeurIPS as work on regression trees has been appearing at NeurIPS and other related conferences for several decades. Yes, others are likely to build on this work and it is a clear improvement over the prior work on TAO.

Originality: I think this is the major weak point of this submission. My main qualm is whether the particular contribution of this submission warrants a separate NeurIPS (which is supposed to be one of the top venues) publication. Ideally, if the authors of the ICML 2020 paper had noticed this particular improvement in the work they had submitted then, it would have merited maybe only a paragraph discussion in that original paper with the rest of it pushed to the Appendix. Also, I believe the particular idea of using the Sherman-Morrison-Woodbury formula to speed up regression has cropped up in many other prior papers. But I agree that it is not something which the TAO paper used so this is novel when restricted to the TAO algorithm.

---

> ### Author Rebuttal · Authors · 2025-07-31
>
> Thanks for your clear review.
>
> **Originality**: our use of the SMW formula may seem obvious in hindsight, but: 1) the authors of the ICML 2020 paper did not realize this for the TAO algorithm, as you note; 2) the many researchers working since the 1980s on the case of CART linear trees did not realize either; and 3) as noted by reviewer 5aDC, we provide a "nontrivial new insight that yields a significant practical gain", since the speedup is large and doesn't involve any approximation, for both TAO or CART. We have added results to speed up CART linear trees in the table in the reply to reviewers 5aDC.
>
> Also, our paper has a second contribution: the non-monotonic behavior of the training complexity predicted by our analysis, very unusual in machine learning models. As reviewer 5aDC notes: "Surprisingly, making the tree deeper beyond can reduce total training time, even though the model has more parameters".

---

> > ### Comment · Reviewer_UDF8 · 2025-08-05
> >
> > I agree with the authors response that although it may seem obvious in hindsight, this paper's use of the SMW formula to speed up TAO was clearly not obvious to the authors of TAO... I am okay with increasing my score.

---

> > > ### Author Response · Authors · 2025-08-08
> > >
> > > We thank the reviewer for increasing their score and for the constructive comments and discussion. Please let us know if there are any further points that remain unaddressed.

---

### Decision · Program_Chairs · 2025-09-17

**Decision:**

Accept (poster)

**Comment:**

The paper proposes a faster training algorithm for regression trees with linear predictors in the leaves, based on the TAO (tree alternating optimization) framework. The enhancement is based on optimizing the leaves' models using the Sherman-Morrison-Woodbury formula to reduce the size of the matrix that needs to be inverted. The paper provides experiments demonstrating large speedups on real and synthetic data.

Overall, the paper proposes a clever use of a technique, which leads to meaningful practical speedups. The main weakness is that the technical novelty is somewhat limited, as it is essentially a (nontrivial) modification of TAO.